

# Majorana zero-modes in a dissipative Rashba nanowire

Arnob Kumar Ghosh$^\star$ and Annica M. Black-Schaffer$^\dagger$

Department of Physics and Astronomy, Uppsala University, Box 516, 75120 Uppsala, Sweden

$\star$ arnob.ghosh@physics.uu.se , $\dagger$ annica.black-schaffer@physics.uu.se

## Abstract

Condensed matter systems are continuously subjected to dissipation, which often has adverse effects on quantum phenomena. We focus on the impact of dissipation on a superconducting Rashba nanowire. We reveal that the system can still host Majorana zero-modes (MZMs) with a finite lifetime in the presence of dissipation. Most interestingly, dissipation can also generate two kinds of dissipative boundary states: four robust zero-modes (RZMs) and two MZMs, in the regime where the non-dissipative system is topologically trivial. The MZMs appear via bulk gap closing and are topologically characterized by a winding number. The RZMs are not associated with any bulk states and possess no winding number, but their emergence is instead tied to exceptional points. Further, we confirm the stability of the dissipation-induced RZMs and MZMs in the presence of random disorder. Our study paves the way for both realizing and stabilizing MZMs in an experimental setup, driven by dissipation.

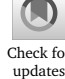

# 1   Introduction

The quest for Majorana fermions in topological superconductors has been attracting an enormous amount of interest for the past two decades [1–7], especially since they are proposed to be an essential ingredient of topological quantum computers [8–12]. The pioneering paper by Kitaev proposes that Majorana fermions can appear as Majorana zero-modes (MZMs) in a spinless $p$-wave superconductor [1]. However, the lack of a suitable candidate for a spinless $p$-wave superconductor barricades the experimental realizability of this model. Nevertheless, Kitaev-like physics has been shown to be possible in several different experimentally realizable setups. One of the most popular routes is based on a one-dimensional (1D) nanowire with Rashba spin-orbit coupling (SOC) in proximity to a conventional spin-singlet $s$-wave superconductor and an applied magnetic field [3,4,6,13–16]. Numerous experiments have been performed based on this Rashba nanowire-superconductor (NW-SC) heterostructure, primarily employing transport measurements to find signature of the MZMs [15–27]. However, to date, a smoking-gun signature of the MZMs is yet to be observed, especially as quantum dots-induced Andreev bound states or the Kondo effect due to magnetic impurities present in the system have been shown to easily mimic MZM signals in transport-based experiments [28–36].

In addition to the issues above for finding MZMs, topological systems are typically described from the viewpoint that they are not coupled to the environment, i.e. they are considered as entirely isolated systems. However, in reality, they are never truly isolated; they constantly interact with their surroundings as open systems [37,38]. In an open system, the long-time evolution is determined by the density matrix. Under the Markovian approximation i.e., when the system is coupled to a memoryless bath [37,39], the evolution of the density matrix is governed by a Liouvillian superoperator that takes the form of the Gorini-Kossakowski-Sudarshan-Lindblad master equation, or simply the Lindblad master equation [40,41]. Although the application of the Lindblad master equation to topological systems has so far been somewhat limited, it has still been used to study dissipation-engineered preparation of some topological non-equilibrium steady states, which have also been proposed for topological quantum computations [39,42–49]. The application of the Lindblad master equation is further not limited to condensed matter systems but has been extensively employed e.g., in quantum information [50], quantum optics [51], and atomic physics [52].

The past few years have also seen a huge interest in non-Hermitian (NH) physics, which can encode some aspects of open systems. Especially the study of NH topological systems has been blooming owing to their intriguing properties that are absent in their Hermitian counterparts, such as the NH skin effect [53–57], the emergence of exceptional points (EPs) [55–63], breakdown of conventional bulk-boundary-correspondence [55,56,64,65], and ramification of symmetry operators [66]. NH open systems have also been shown to sometimes harbor boundary states that are stabilized in the presence of dissipation [67–71]. In this context, the Liouvillian superoperator that drives the dynamics of the density matrix of the open system is an NH matrix, which then unravels the underlying NH topology [72–82].

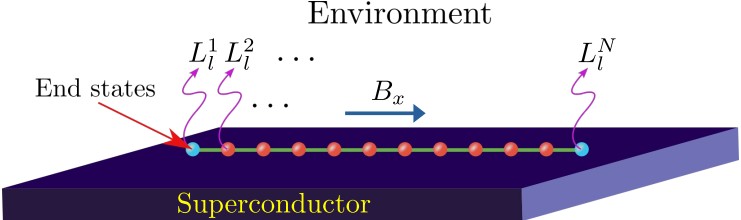

Figure 1: Schematic representation of NW-SC heterostructure setup encompassing a 1D Rashba NW (orange dots connected by green lines) in close proximity to a bulk conventional spin-singlet $s$-wave superconductor (violet) with an applied in-plane magnetic field $B_x$. The system can host MZMs (cyan dots) in the topological regime. The heterostructure is coupled to the environment via dissipation. The loss due to the environment is encoded through the jump operators: $L_l^1, L_l^2, \cdots, L_l^N$; with $N$ being the total number of lattice sites.

Furthermore, the interplay of NH physics and superconductivity has recently exposed several intriguing phenomena to the above examples, such as the generation of odd-frequency pairing [83], emergence of NH Bogoliubov Fermi arcs [84,85], enhancement of superconductivity by EPs [86], and topological phase transition with infinitesimal instability [87]. It has also been established that MZMs can be obtained in a NH topological superconductor [86–93], satisfying a modified NH Majonara condition [87]. Recently, it has also been argued that non-Hermiticity can resolve the Majorana vs. Andreev controversy [94]. Thus, non-Hermiticty provides a very interesting avenue for topological superconductivity. However, most of the exploration of topological superconductors within an NH setting is done by utilizing only some specific forms of NH terms, often even just added ad-hoc. A few counterexamples exist, such as creating an NH topological superconductor by coupling the superconductor to a ferromagnetic lead [83–85,88] or allowing dissipation in the system within the Kitaev model [68,95].

In particular, to date, no systematic study exists of the effect of dissipation, thereby creating an unexplored NH regime, on more experimentally realizable models for topological superconductivity, such as Rashba NW-SC heterostructures [3,4,6,13–16]. In fact several intriguing questions exist: (a) What is the effect of dissipation on the Rashba NW-SC heterostructure, and do we continue to obtain MZMs in such a dissipative system? (b) Can dissipation even lead to the generation of topological superconductivity and MZMs when starting from a Hermitian non-topological regime? In this work, we answer both of these fascinating questions.

In particular, we consider a 1D NW with Rashba SOC in direct proximity to a conventional spin-singlet $s$-wave superconductor and with an applied in-plane magnetic field, which is exposed explicitly to the surrounding environment, see Fig. 1. We employ the Lindblad master equation to obtain an NH description of this setup in terms of Lindblad spectra. First, we explore the fate of MZMs, present in the Hermitian (closed system) limit, for increasing dissipation. We show that the MZMs survive in the dissipative environment but obtain a finite lifetime, see Fig. 2. Next, we demonstrate that dissipation can even induce in-gap states localized at the edges of the system, where the system in the Hermitian is non-topological and with no in-gap states. Here, we obtain two kinds of zero-modes (ZMs): four robust ZMs (RZMs), whose origin we trace back to the presence of EPs, and two ZMs mimicking MZMs, originating at a topological phase transition where the bulk gap closes and a finite topological winding number is achieved (see Figs. 3 and 4). Notably, we find that both the RZMs and MZMs are robust against disorder (see Figs. 5 and 6). Our study uncovers intriguing aspects of dissipative systems, where we observe that dissipation goes hand-in-hand with NH physics and even renders the generation of topologically protected MZMs.

The remainder of this work is organized as follows. In Section 2, we discuss the model Hamiltonian, introduce the form of dissipation, and review the method that we employ. Section 3 presents the main results. In particular, we discuss the fate of MZMs under uniform dissipation that are present already in the Hermitian topological regime in Section 3.1, while we explore the generation of RZMs and MZMs, as well as their stability against disorder in Section 3.2. Finally, we conclude the work with the conclusions and outlook in Section 4. In Appendix A and Appendix B, we present more details about the method, while in Appendix C, we discuss the effect of single-site dissipation on our system, which supplements our results in the main text.

## 2 Model and method

In this section, we discuss the details of the system, its Hamiltonian, the nature of the dissipation, and the methods to handle this system.

### 2.1 Rashba NW-SC heterostructure

We consider a 1D NW with Rashba SOC placed on top of a conventional spin-singlet $s$-wave superconductor with an applied in-plane magnetic field. The schematic representation of this setup is depicted in Fig. 1. The system under consideration can be described by the following Hamiltonian [3, 4, 6, 13–16]

$$H = -\mu \sum_{i=1,\alpha}^{N} c_{i\alpha}^{\dagger} c_{i\alpha} + \left( t_{\mathrm{h}} \sum_{i=1,\alpha}^{N-1} c_{i\alpha}^{\dagger} c_{i+1\alpha} + \mathrm{H.c.} \right) - \left( i\lambda_{\mathrm{R}} \sum_{i=1,\alpha\beta}^{N-1} c_{i\alpha}^{\dagger} (\sigma_z)_{\alpha\beta} c_{i+1\beta} + \mathrm{H.c.} \right)$$
$$+ B_x \sum_{i=1,\alpha\beta}^{N} c_{i\alpha}^{\dagger} (\sigma_x)_{\alpha\beta} c_{i\beta} + \left( \Delta \sum_{i=1}^{N} c_{i\uparrow}^{\dagger} c_{i\downarrow}^{\dagger} + \mathrm{H.c.} \right), \tag{1}$$

where $\mu$, $t_{\mathrm{h}}$, $\lambda_{\mathrm{R}}$, $B_x$, and $\Delta$ represent the chemical potential, hopping amplitude, strength of the Rashba SOC, magnitude of the in-plane magnetic field, and the proximity-induced $s$-wave superconducting order parameter, respectively. Here, $c_{i\alpha}$ ($c_{i\alpha}^{\dagger}$) stands for the electron creation (annihilation) operator at lattice site $i$ with spin $\alpha = \uparrow, \downarrow$, and $N$ denotes the number of lattice sites in the 1D wire. The Pauli matrices $\boldsymbol{\sigma}$ acts on the spin degrees of freedom. For a NW of infinite length and thus translationally invariant, we obtain a momentum-space Bloch form of the Hamiltonian in Eq. (1) in the Bogoliubov-de Gennes (BdG) basis $\Psi_k = \left\{ c_{k\uparrow}, c_{k\downarrow}, -c_{-k\downarrow}^{\dagger}, c_{-k\uparrow}^{\dagger} \right\}$ reading $H_{\mathrm{BdG}} = \frac{1}{2} \sum_k \Psi_k^{\dagger} H(k) \Psi_k$, with $H(k)$

$$H(k) = (-\mu + 2t_{\mathrm{h}} \cos k)\tau_z \sigma_0 + 2\lambda_{\mathrm{R}} \sin k \tau_z \sigma_z + B_x \tau_0 \sigma_x + \Delta \tau_x \sigma_0, \tag{2}$$

where the Pauli matrices $\boldsymbol{\tau}$ acts on the particle-hole (PH) degrees of freedom. The Hamiltonian $H(k)$ respects both PH symmetry $\mathcal{C} = \tau_y \sigma_y K$: $\mathcal{C}^{-1} H(k) \mathcal{C} = -H(-k)$ with $K$ being the complex-conjugation operator and chiral symmetry $S = \tau_y \sigma_z$: $SH(k)S = -H(k)$. However, $H(k)$ breaks the time-reversal symmetry $\mathcal{T} = \tau_0 \sigma_y K$: $\mathcal{T}^{-1} H(k) \mathcal{T} \neq H(-k)$ due to the presence of the nonzero magnetic field $B_x$. We obtain the critical magnetic field needed a topological phase transition by investigating the bulk gap-closing. At $k = 0$ and $\pi$, the Hamiltonian $H(k)$, Eq. (2), exhibits gap-closing for $B_x = |B_{x,c1}| = \sqrt{(\mu - 2t_{\mathrm{h}})^2 + \Delta^2}$ and $B_x = |B_{x,c2}| = \sqrt{(\mu + 2t_{\mathrm{h}})^2 + \Delta^2}$, respectively [13, 96]. The system hosts two MZMs localized at the wire endpoints (one MZM per endpoint) in the topological phase existing for $|B_{x,c1}| < B_x < |B_{x,c2}|$. Throughout this work, we for simplicity choose $t_{\mathrm{h}} = \mu = \Delta = 1.0$ and $\lambda_{\mathrm{R}} = 0.5$, unless mentioned otherwise. Our observations do not explicitly depend on this specific choice of parameters and will remain qualitatively the same for different choices of parameters.

## 2.2 Lindblad master equation and dissipation

We consider that the system in Fig. 1 is coupled to an outside environment or bath, and thereby subject to dissipation. We employ the Markovian approximation, such that the environment is memoryless. This approximation is valid when the dynamics of the environment are faster than the system. In this case the evolution of the density matrix $\rho(t)$ follows the Lindblad master equation [37, 39–41]:

$$\frac{\partial \rho(t)}{\partial t} = -i[H, \rho(t)] - \frac{1}{2}\sum_m \left(\left\{L_m^\dagger L_m, \rho(t)\right\} - 2L_m^\dagger \rho(t)L_m^\dagger\right), \tag{3a}$$

$$\equiv \mathcal{L}\rho(t), \tag{3b}$$

where $\mathcal{L}$ is the Liouvillian superoperator and $L_m$'s are the Lindblad jump operators. The effect of the bath-system coupling is encoded in these jump operators [37, 39]. The first term on the right-hand side of Eq. (3a) resembles the Liouville-von Neumann equation of motion and represents the unitary evolution of the system in the absence of dissipation. The second term is divided into two parts: the first part describes the continuous loss of energy to the environment, and the second part denotes quantum jumps [74]. Without the quantum jumps, i.e. in the semi-classical limit, we obtain non-unitary evolution of the density matrix $\rho(t)$, governed by an effective NH Hamiltonian $H_{\text{eff}} = H - \frac{i}{2}\sum_m L_m^\dagger L_m$, which also represents the short-time dynamics of the system. In our work, we go beyond such an effective NH Hamiltonian description and use the full formalism, including the effects of the quantum jumps. This adds substantial complexity to solving the problem, as we discuss below. Furthermore, quantum jumps have also been observed in experiments [97–101], and thus, incorporating the effect of quantum jumps is necessary for open systems.

The jump operators $L_m$ can represent both onsite and bond losses or gains. For simplicity, we consider a simple form of onsite loss, which can become experimentally feasible. Although we may choose the form of the jump operators differently, our work establishes that even this simple choice of jump operators leads to interesting findings. The chosen jump operator at lattice site $i$ reads

$$L_l^i = \sqrt{\gamma_i}\left(c_{i\uparrow} + c_{i\downarrow}\right), \tag{4}$$

where the index $l$ stands for loss, and $\gamma \geq 0$ encapsulates the strength of the loss to the environment. In the main text, we consider uniform loss i.e., $L_l^i \neq 0 \ \forall i$ and $\gamma_i = \gamma \ \forall i$, while in Appendix C, we provide complementary results for single site loss.

### 2.2.1 Lindblad spectrum

To incorporate the full effects of openness within the Lindblad formalism, including the effects of quantum jumps, we employ third quantization. The details of the third quantization method are introduced in Refs. [102–104]. For the sake of completeness, we provide the main steps for obtaining the matrix form of the Liouvillian $\mathcal{L}$ here, while the technical details are found in Appendix A. To proceed, we represent all the complex fermions in terms of real Majorana fermions by employing the transformation

$$c_{i\uparrow} = \frac{(w_{Ai} + iw_{Bi})}{2} \quad \text{and} \quad c_{i\downarrow} = \frac{(w_{Ci} + iw_{Di})}{2}, \tag{5}$$

where $w_{\alpha i}$ represent real Majorana fermions, satisfying the fermionic anti-commutation relation as $\{w_{\alpha i}, w_{\beta i}\} = 2\delta_{\alpha,\beta}\delta_{i,j}$; with $\alpha, \beta = A, B, C, D$ and $i, j = 1, 2, \cdots, N$. We recast the

Hamiltonian Eq. (1) and the jump operator, Eq. (4) in the Majorana basis as

$$H_{\mathrm{M}} = \sum_{a,b}^{4N} w_a H_{a,b} w_b \,, \tag{6}$$

$$L_l = \sum_{b}^{4N} l_{l,b} w_b \,, \tag{7}$$

where $H_{\mathrm{M}}$ is a $(4N \times 4N)$ matrix and $L_l$ is $(4N \times 1)$ column matrix. Here, we define the vector $\underline{w} = \{w_{A1}, w_{B1}, w_{C1}, w_{D1}, \cdots, w_{Ai}, w_{Bi}, w_{Ci}, w_{Di}, \cdots\}^T$, with $T$ being the transpose operation. The next step is to consider a Fock-space $\mathcal{K}$, also known as Liouville-Fock space. The density matrix is part of that space $\mathcal{K}$, $\rho(t) \in \mathcal{K}$, and is a $(16N^2 \times 1)$-vector in this Liouville-Fock space (while $\rho(t)$ is a $(4N \times 4N)$ matrix in the normal Hilbert space). We consider the vector space $\mathcal{K}$ spanned by the polynomial $P_{\underline{\alpha}}$, such that

$$P_{\underline{\alpha}} = \prod_{a=1}^{4N} w_a^{\alpha_a} \,, \tag{8}$$

where $\alpha_a = 0, 1$ and denotes the occupation of the Majorana fermion states. Thus, the inner product is defined as $\langle x|y \rangle = 2^{-4N} \mathrm{tr}\left[x^\dagger y\right]$. We also define the adjoint fermion annihilation and creation operator as

$$\phi_a |P_{\underline{\alpha}}\rangle = \delta_{\alpha_a,1} |w_a P_{\underline{\alpha}}\rangle \quad \text{and} \quad \phi_a^\dagger |P_{\underline{\alpha}}\rangle = \delta_{\alpha_a,0} |w_a P_{\underline{\alpha}}\rangle \,. \tag{9}$$

Since, $w_a^2 = 1$ $\phi_a$ annihilates the Majorana fermion $w_a$, while $\phi_a^\dagger$ creates the Majorana fermion $w_a$. These operators follow the fermion anti-commutation relation $\{\phi_a, \phi_b^\dagger\} = \delta_{a,b}$.

Assuming an even number of fermionic degrees of freedom in the system, we can now represent the Liouvillian that governs the dynamics of the dissipative NW-SC heterostructure by $\mathcal{L}_+$. We provide the steps to obtain $\mathcal{L}_+$ in Appendix A and here we only report the expression:

$$\mathcal{L}_+ = \frac{i}{2} \left(\underline{\phi}^\dagger \cdot \quad \underline{\phi} \cdot\right) \begin{pmatrix} -X^\dagger & -iY \\ 0 & X \end{pmatrix} \begin{pmatrix} \underline{\phi}^\dagger \\ \underline{\phi} \end{pmatrix} - A_0 \,, \tag{10}$$

where we define $X = 4H_{\mathrm{M}} + i(M + M^T)$, $Y = 2(M - M^T)$, and $A_0 = \frac{1}{2}\mathrm{Tr}[X]$, with $M_{jk} = l_{j,l}^T l_{l,k}^*$. Here, $\underline{\phi}^\dagger$ and $\underline{\phi}$ represent the vector form of the adjoint creation and annihilation operators, respectively. Since $\mathcal{L}_+$ takes an upper triangular form, the spectrum of the Liouvillian, which is also called the Lindblad spectrum, is completely determined by that of $X$, i.e. it is the operator $X$ that dictates the dynamics of the system [79]. Thus, we obtain

$$\mathcal{L}_+ = i \sum_a E_a \chi_a' \chi_a \,, \tag{11}$$

where $E_a$ are the eigenvalues of $X$ and $\chi_a'$ and $\chi_a$ are the normal-master modes satisfying $\{\chi_a', \chi_b\} = \delta_{a,b}$ [102–104]. It is also known that the topological properties of $\mathcal{L}_+$ are mimicked by the matrix $X$ [73]. Thus, we can focus fully on $X$, as it dictates both the dynamical and topological properties of the system. Owing to the form of the loss in Eq. (4), we can finally obtain the matrix $X$ for a dissipative NW-SC heterostructure, which in momentum-space reads

$$X(k) = 4H_{\mathrm{M}}(k) + i\left(M + M^T\right) \tag{12}$$

$$= (-\mu + 2t_{\mathrm{h}} \cos k)\tilde{\tau}_0 \tilde{\sigma}_y + 2\lambda_{\mathrm{R}} \sin k \tilde{\tau}_z \tilde{\sigma}_0 - B_x \tilde{\tau}_x \tilde{\sigma}_y + \Delta \tilde{\tau}_y \tilde{\sigma}_x + \frac{i\gamma}{2}\left(\tilde{\tau}_0 \tilde{\sigma}_0 + \tilde{\tau}_x \tilde{\sigma}_0\right) \,,$$

where $\tilde{\tau}$ and $\tilde{\sigma}$ are Pauli matrices.

Further, we note that one of the differences between $X$ and the NH Hamiltonians so far usually considered in the literature [55,56] is that the imaginary part of the eigenvalue of the matrix $X$, i.e., the Lindblad spectrum Im $E$, is always greater than zero, which ensures that the density matrix decays over time and approaches its steady state $\rho_{SS}$: $\rho_{SS} = \exp[i\mathcal{L}_+ t]\rho(0)$ as $t$ goes to infinity [103]. While studying the topological properties of the steady state is also important [45, 47], in our work, we focus on the spectral topological properties of the Liouvillian, as these are independent of each other [73] and the Louvillian offers the natural connection to NH physics. In the latter context we note that Im $E \geq 0$ provides a 10-fold classification of the Lindbladians [73], in contrast to all NH Hamiltonians, which exhibit a 38-fold classification instead [66].

## 2.3  Winding number

To investigate the topological properties of a dissipative NW-SC heterostructure, we compute a winding number based on the matrix $X$. The matrix $X(k)$ satisfies pseudo-anti-Hermiticity symmetry ($\Gamma$): $\Gamma X^\dagger(k)\Gamma = -X(k)$; with $\Gamma = \tilde{\tau}_x \tilde{\sigma}_x$ [105]. For an NH system, this symmetry is also called chiral symmetry [66]. We employ this pseudo-anti-Hermiticity symmetry to define the winding number $\nu$ as [106]

$$\nu = \frac{i}{2\pi} \int_{BZ} dk \; \mathrm{tr}\left[q(k)^{-1}\partial_k q(k)\right].$$

(13)

We provide the detailed procedure to obtain $q(k)$ and $\nu$ in Appendix B. Our results establish that we do not observe any skin effects and that the conventional Hermitian bulk-boundary correspondence is intact. Thus, we can always employ this momentum space topological invariant to obtain the topological phase boundaries of the system.

# 3  Results: NW-SC heterostructure under uniform dissipation

This section is devoted to our main results. We consider two cases: first, when the isolated (Hermitian) Hamiltonian of the Rashba NW-SC heterostructure is topological, and then when the isolated system is topologically trivial. In both cases, we mainly focus on the Lindblad spectra i.e., the eigenvalue spectrum of the matrix $X$ as described in Section 2.2.1. Throughout this section, we usually employ open boundary condition (OBC) when computing the Lindblad spectra and the local density of states (LDOS), unless mentioned otherwise. We further find that the boundary modes always have their real energy equal to zero, which makes us term them ZMs, akin to those in the Hermitian system.

## 3.1  Fate of MZMs under uniform dissipation

In a realistic experimental scenario, a setup is always in contact with the environment. Thus, it is worth examining the fate of MZMs in a topological 1D Rashba NW-SC heterostructure. To this end, we consider the isolated setup, i.e. system Eq. (1) with no dissipation, to reside in the topological regime: $|B_{x,c1}| < B_x < |B_{x,c2}|$ [13,96]. We then add uniform dissipation and we plot the real part of the eigenvalue spectra Re $E$ of the matrix $X$ as a function of the dissipation strength $\gamma$ in Fig. 2(a). The Re $E$ are further colored corresponding to their imaginary part Im $E$. As seen from the color bar of Fig. 2(a), Im $E \geq 0$ always, ensuring the decay of the density matrix over time. Notably, the MZMs present in the Hermitian limit, $\gamma = 0$, continue to exist as Re $E = 0$ states, but they now carry also a finite imaginary part, which increases with increasing $\gamma$. This finite imaginary part can be understood as the lifetime of the MZMs in a dissipative system, and as such, they would appear as a broadened peak in, e.g. spectroscopy

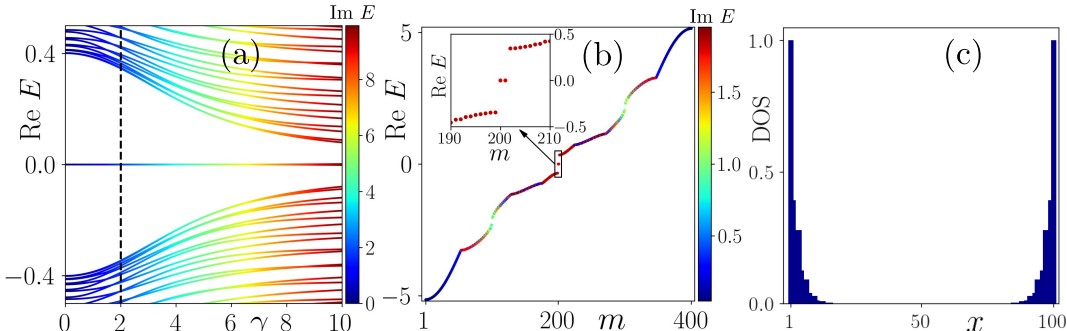

Figure 2: (a) Real part of the Lindblad spectra Re $E$ as a function of dissipation strength $\gamma$. (b) Re $E$ of the Lindblad spectrum as a function of the state index $m$, at the black dashed line in (a). Inset shows zoomed-in eigenvalue spectrum close to Re $E = 0$. The Re $E$ is weighted by the Im $E$ value, see color bar in (a,b). (c) Local density of states (LDOS) computed for Re $E = 0$ states as a function of lattice sites $x$ at the black dashed line in (a). Here $B_x = 2.0 > |B_{x,c1}|$ and we use 100 lattice sites.

measurements. It is here also worth mentioning that, while the MZMs carry a reasonably high value of imaginary parts, they are not the states with the maximum imaginary parts. Furthermore, we find that the gap protecting the MZMs from all other states diminishes with $\gamma$, especially for large dissipation, as clearly depicted in Fig. 2(a).

We next choose a small realistic value of the dissipation strength (black dashed line in Fig. 2(a)) and examine the Lindblad spectrum and the LDOS. In Fig. 2(b), we plot the real part of the Lindblad spectrum Re $E$ as a function of the state index $m$. The color again represents the imaginary part of the eigenvalue Re $E$. In the inset, we show the states closest to Re $E = 0$, which corroborates the presence of two MZMs. The LDOS associated with these two MZMs is illustrated in Fig. 2(c). We clearly see that the MZMs are localized at the end of the NW, even for the dissipative system. We further confirm the topological properties of the system by computing the winding number $\nu$ using Eq. (13). We always obtain $\nu = 1$ when MZMs are present in the system. These results show that the MZMs present in the non-dissipative, or Hermitian, topological regime of a 1D Rashba NW-SC heterostructure survive in a dissipative background. However, the MZMs now also carry finite imaginary parts and thus can survive only up to the time scale set by the dissipation strength $\gamma$. This constraint applies to all types of ZMs we obtain in this work. In Appendix C, we establish the same results for single-site dissipation.

## 3.2 Emergence of MZMs and RZMs under dissipation

Having established that the MZMs in the Hermitian Rashba NW-SC heterostructure survive under dissipation, we next investigate the effect of dissipation in the topologically trivial limit and particularly seek whether we can obtain non-trivial topology with boundary states via dissipation only. To achieve this, we set $B_x < |B_{x,c1}|$, such that the isolated NW-SC heterostructure does not exhibit any zero-energy end states and where the bulk is always trivially gapped with $\nu = 0$. Then, we couple this system with the environment and illustrate the results in Figs. 3, 4, and 5. We uncover the generation of two kinds of ZMs: four robust ZMs (RZMs), which we will show are not connected to the bulk topology, and two MZMs, which emerge due to the closing of the bulk gap and finite bulk winding number.

First, we investigate the phase diagram as a function of $\gamma$. We plot Re $E$ as a function of dissipation strength $\gamma$ in Fig. 3(a) and observe that there are no states at Re $E = 0$ when $\gamma = 0$ but with the increase in $\gamma$, we observe the appearance of states at or near Re $E = 0$, see yellow

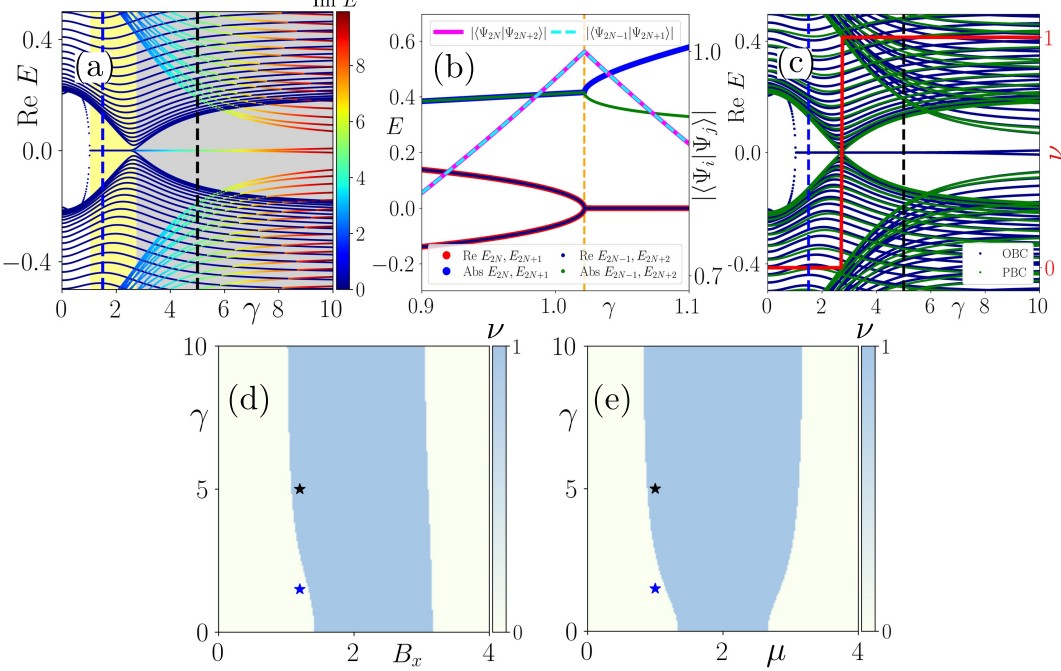

Figure 3: (a) Real part of the Lindblad spectra Re $E$ as a function of dissipation strength $\gamma$, with color bar representing the imaginary part of the eigenvalue. (b) Eigenvalues of the four middle states: $(2N-1)$, $2N$, $(2N+1)$, and $(2N+2)$, i.e. closest to Re $E = 0$. Right axis corresponds to the scalar product $|\langle \Psi_i | \Psi_j \rangle|$ of the corresponding eigenstates. Orange dashed line indicates the position of EPs. (c) Real part of the Lindblad spectra employing OBC (blue dots) and PBC (green dots) as a function of $\gamma$ and winding number $\nu$ on the right axis (red line). The winding number exhibits a jump from 0 to 1 when there is a gap closing in the PBC spectrum. (d,e) Winding number $\nu$ in the $B_x$-$\gamma$ plane for fixed $\mu = 1.0$ in (d) and in the $\mu$-$\gamma$ plane for fixed $B_x = 1.2$ (e). Color stars correspond to the dashed vertical lines in (a,c). Here $B_x = 1.2 < |B_{x,c1}|$ in panels (a,b,c).

and gray regions in Fig. 3(a). In the yellow and gray regions, we obtain four RZMs and two MZMs at Re $E = 0$, respectively. The yellow region emerges without any closing of the bulk gap, see the left part of the yellow region in Fig. 3(a) and also the eigenvalue spectra for the system obeying periodic boundary condition (PBC) in Fig. 3(c). Nevertheless, we observe that the left edge of the yellow region in Fig. 3(a) corresponds to EPs. In Fig. 3(b), we illustrate this formation of two second-order EPs: the $2N$- and $(2N+2)$-th, and the $(2N-1)$- and $(2N+1)$-th states coalesce pairwise with each other, forming two second-order EPs at the orange dashed line (corresponding to the left edge of the yellow region). Notice that the real, imaginary, and absolute eigenvalues all merge at the orange dashed line in Fig. 3(b). At this point, the scalar product $|\langle \Psi_{2N} | \Psi_{2N+2} \rangle|$ (magenta solid line) and $|\langle \Psi_{2N-1} | \Psi_{2N+1} \rangle|$ (cyan dashed line) also becomes unity, thus forming two parallel set of states, another signature of EPs. EPs are known to be topological [55–63] and we here find that EPs can additionally also induce ZMs. This finding of EP-generated ZMs, here called RZMs due to their disorder stability, as found below, is one of the unexpected findings of this work. We further note that the RMZs appearing in the yellow region cannot possess a finite winding number, since they are not connected to any bulk states. Interestingly, similar features have recently been observed in the Su-Schrieffer-Heeger model, where the emergence of ZMs was also not found to be associated with a bulk gap closing [67, 107].

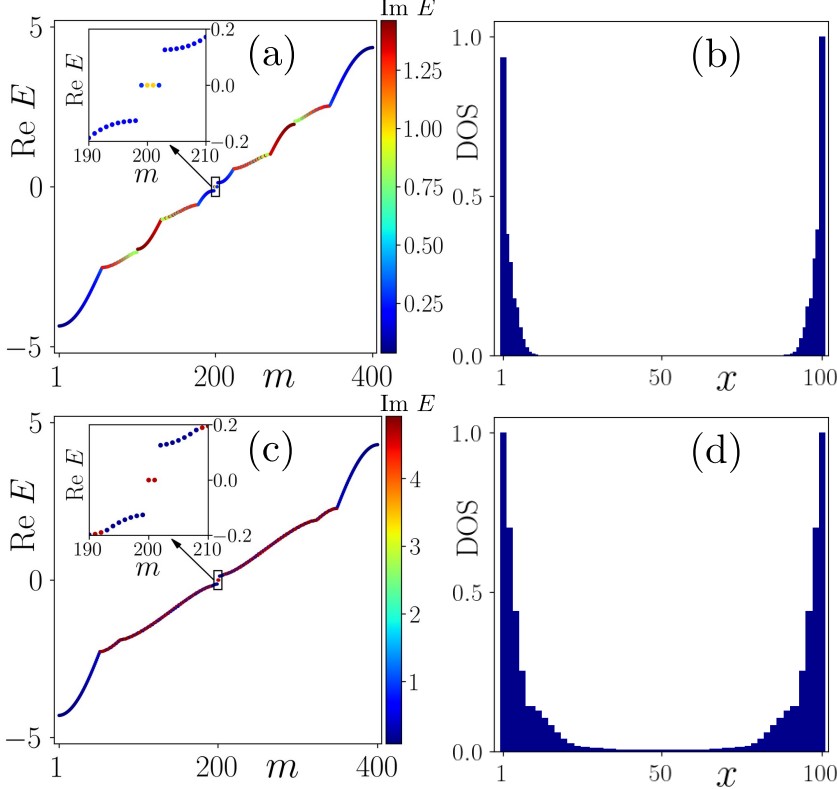

Figure 4: (a) [(c)] Real part of the Lindblad spectrum Re $E$ as a function of state index $m$ for $\gamma = 1.5$ [$\gamma = 5.0$]. Modes close to Re $E = 0$ are shown in the inset. Color bar represents Im $E$. (b,d) LDOS computed for the Re $E = 0$ states in (a,c). Here $B_x = 1.2 < |B_{x,c1}|$.

Next, we focus on the gray region of Fig. 3(a), which occurs after a clear bulk gap closing at the interface between the yellow and the gray region. In this phase, we obtain two zero-energy modes, designated as MZMs below (see Fig. 4(c)). The bulk gap closing is clearly seen in the PBC eigenvalue spectra (green dots) in Fig. 3(c). Since the MZMs emerging in this phase are connected to the bulk states, we can compute the winding number $\nu$, thereby employing the bulk states to unravel the bulk topology. We depict the winding number $\nu$ as a function of $\gamma$ on the right axis of Fig. 3(c). The winding number jumps from 0 to 1 at the bulk gap closing point, thus exhibiting a non-zero value as soon as the MZMs are present in the system. We thus find a topological phase transition exhibiting MZMs when the bulk gap closes. This non-trivial phase with $\nu = 1$ is clearly driven by dissipation and is not present in the closed system. This is the second unexpected finding of this work. To map out the phase diagram, we also plot the winding number $\nu$ in the $B_x$-$\gamma$ and $\mu$-$\gamma$ planes in Fig. 3(d) and (e), respectively. These phase diagrams show that dissipation can drive a trivial NW-SC heterostructure with $\nu = 0$ in the Hermitian limit ($\gamma = 0$) to a topologically non-trivial phase with $\nu = 1$ with dissipation-induced MZMs. We observe a visible increase in the parameter space for $\gamma \neq 0$ compared to $\gamma = 0$, for example, as indicated by the colored stars corresponding to the vertical dashed lines in Fig. 3(a,c). This means we can obtain MZMs for a smaller value of the magnetic field $B_x \leq |B_{x,c1}|$ for a fixed $\mu$, if we allow for dissipation and we have dissipation-induced MZMs. We finally note that, in Fig. 3(a,c), the splitting of the MZMs for higher values of dissipation strength $\gamma$ occurs due to finite overlap of the wavefunctions of the MZMs. As such this splitting disappears for longer NWs.

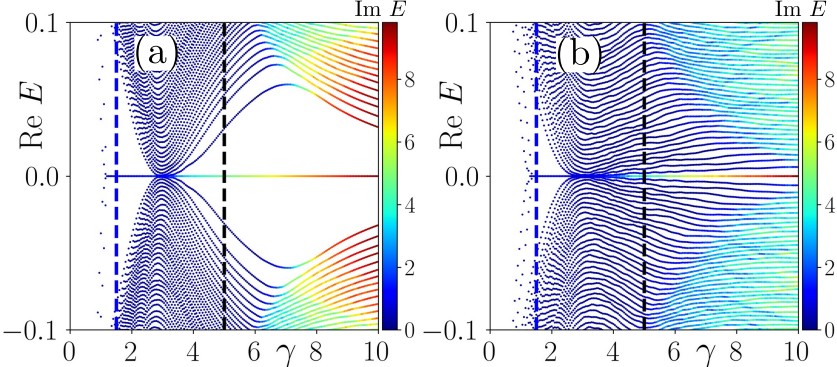

Figure 5: Disorder-averaged real part of the Lindblad spectra Re $E$ as a function of dissipation strength $\gamma$ for $w = 0.5$ (a) and $w = 1.0$ (b). Color bar represents Im $E$. Here $B_x = 1.2 < |B_{x,c1}|$, and we use 500 lattice sites.

Having understood the emergence of RZMs and MZMs from the phase diagram, we now further investigate their properties. In particular, we show the real part of the Lindblad spectrum Re $E$ corresponding to the blue dashed line in Figs. 3(a,c) (or equivalently blue stars in Figs. 3(d,e)), as a function of the state index $m$ in Fig. 4(a). In the inset of Fig. 4(a), we demonstrate the eigenvalues close to Re $E = 0$, which corroborates the presence of four RZMs. However, they have different imaginary parts, divided into two sets: orange and blue. This can be interpreted as different lifetimes of the RZMs. To study their localization properties, we plot the LDOS computed at Re $E = 0$ in Fig. 4(b), which indicates that the RZMs are localized at the two ends of the system. This is similar to any topological boundary state, despite the RZMs not appearing in a topologically non-trivial phase, as indicated by the zero winding number. Furthermore, we observe that the RZMs with degenerate imaginary parts are localized at two opposite ends.

In Fig. 4(c), we depict Re $E$ as a function of the state index $m$ corresponding to the black dashed line in Figs. 3(a,c) (or equivalently black stars in Figs. 3(d,e)). We observe from the inset that there are now two states at Re $E = 0$, the two MZMs already identified Fig. 2(b). Here, both the Re $E = 0$ MZMs incur the same value of imaginary part i.e., they have the same lifetime. We illustrate the LDOS associated with the MZMs at Re $E = 0$ in Fig. 4(d), showing how they are primarily localized to the end points of the wire. We also note that the peak height of the two MZMs are equal in Fig. 4(d), in contrast with the RZMs in Fig. 4(b), where the peak heights are not the same in both the ends. Taken together, these results illustrate that dissipation alone can lead to the generation of ZMs, both topologically non-trivial MZMs, and topologically trivial RZMs, in an otherwise trivial system with no in-gap states. Furthermore, we also investigate a spin-polarized version of the loss operator (Eq. (4)), i.e., we consider different dissipation strengths for the two spin species. However, we do not observe any dramatic changes to our results as long as the spin polarization is not complete. In particular, we still obtain both the dissipation-induced MZMs and RZMs.

### 3.2.1 Stability against disorder

Having demonstrated the emergence of both RZMs and the MZMs in a dissipative NW with no topology in the closed Hermitian limit, we check the stability of these states in the presence of disorder. This is a particularly valid question for the RZMs, since they do not enjoy any protection from the bulk topology, but elucidating disorder stability also for MZMs is of experimental relevance. To this end, we add an onsite, Anderson-type, non-magnetic random disorder potential to the Hamiltonian of the NW: $-\sum_{i\alpha} V_i c_{i\alpha}^{\dagger} c_{i\alpha}$. Here $V_i$ is uniformly distributed in the range $[-w/2, w/2]$, with $w$ the strength of the disorder potential.

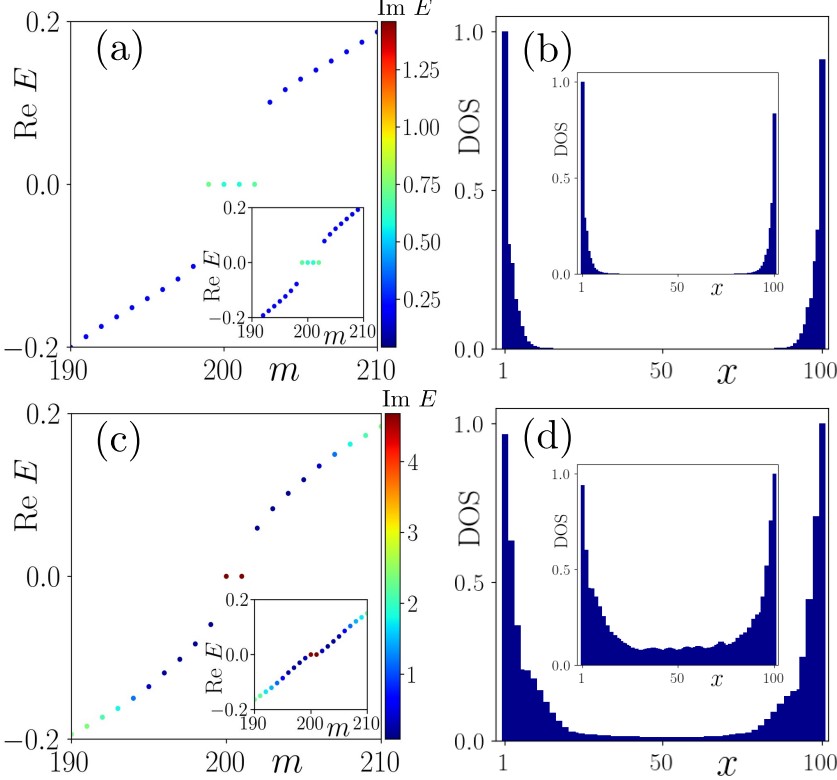

Figure 6: (a) Disorder-averaged real part of the Lindblad spectrum Re $E$ as a function of state index $m$ and (b) the LDOS computed for Re E = 0 for disorder strength $w = 0.5$. In the inset of panels (a) and (b), we repeat (a) and (b) for $w = 1.0$. Here $\gamma = 1.5$. In panels (c) and (d), we repeat (a) and (b) for $\gamma = 5.0$. Here $B_x = 1.2 < |B_{x,c1}|$.

We consider disorder averages over 50 and 500 random configurations of the disorder potential to obtain the disorder-average Lindblad spectra in Figs. 5 and 6, respectively, and also check for convergence in disorder sampling. To this end, we sort the eigenvalues according to their real parts in ascending order for each random disorder realization. If two or more eigenvalues have the same real parts, we also sort their corresponding imaginary parts in ascending order. We show the disorder-averaged real part of the Lindblad spectra Re $E$ as a function of $\gamma$ in Figs. 5(a) and (b) for $w = 0.5$ and 1.0, respectively. Although disorder-averaged eigenvalues are not directly observable, these spectra still capture the overall effects of disorder, and especially we get to know when disorder-induced (near) ZMs appear as a function of $\gamma$, which signals disorder destroying the clean limit results. For the smaller disorder strength $w = 0.5$ in Fig. 5(a), we do not observe any significant difference compared to the clean case in Fig. 3(a), except that the point separating two phases with the RZMs and the MZMs, respectively, is somewhat broadened in the disordered system. However, the bulk gap in terms of its real energy is reduced in the disorder system. For the comparatively strong disorder strength $w = 1.0$ in Fig. 5(b) (note $t = 1$), we note that a dramatic reduction and even destruction of the real energy bulk gap, such that no bulk gap longer protects the MZMs. Interestingly, the RZMs seem to be more resilient compared to the MZMs, as they remain well separated from all other states even at these strong disorder levels.

To gain more insights into the stability of the RZMs and the MZMs, we select two fixed dissipation strengths, dashed blue and black lines in Fig. 5, and investigate the localization properties of the in-gap states in the presence of disorder. In particular, in Fig. 6(a), we show

the disorder-averaged Re $E$ as a function of state index $m$ for the states close to Re $E = 0$ when the system hosts RZMs and at $w = 0.5$ (blue dashed line in Fig. 5(a)). In the inset, we instead use disorder strength $w = 1.0$ (blue dashed line in Fig. 5(b)). We observe that the RZMs are separated from all other bulk states in terms of their real energy and continue to appear at Re $E = 0$ even for a very disordered NW. In Fig. 5(b), we depict the LDOS associated with the Re $E = 0$ states. The end states continue to be well separated from each other in space. The main conclusion from Fig. 5(b) (with $w = 0.5$) and its inset (with $w = 1.0$) is that with an increase in the strength of the disorder, the RZMs appear to be largely unaltered, only their peak height becomes slightly more asymmetric. Due to this robustness, we name these four ZMs, RZMs.

On the other hand, for the MZMs, the diminishing of the gap separating the zero-energy states and the bulk states is more abrupt with the increase in disorder. We depict the disorder-averaged Re $E$ as a function of the state index $m$ for the states close to Re $E = 0$ in Fig. 5(c) for $w = 0.5$, while using $w = 1.0$ in the inset. For $w = 1.0$, the gap in the real energy is essentially nonexistent. In Fig. 5(d), we investigate the LDOS computed at Re $E = 0$ for $w = 0.5$, while $w = 1.0$ in the inset. The leaking of the MZMs into the bulk is clearly visible for both disorder strengths. This is expected as the MZMs are energetically not very separated from the bulk states anymore and shows how the destruction of the bulk gap directly reduced the MZM localization. Here, we also notice that the peak height at both NW ends is not the same anymore. It is also interesting to compare the stability towards the disorder of these dissipation-induced MZMs to their Hermitian topological counterpart, having the same gap in the real part of the eigenvalue. We find that the MZMs appearing in the Hermitian topological case are overall more robust compared to the dissipation-induced ones. This phenomenon can be understood from the fact that the diminishing of the bulk gap with the disorder is much less in the Hermitian counterpart. In particular, we can still obtain MZMs separated from the other states in the Hermitian case for $w = 1$.

We also investigate the effect of non-uniform dissipation strength at each lattice site. In particular, we consider the case where the dissipation strength $\gamma_i = \gamma + \delta\gamma_i$ with $\delta\gamma_i$ randomly distributed in $[-W/2, W/2]$ and $W$ being the strength of the dissipation variation (or disorder). However, as long as $W$ is small, we do not observe any substantial changes to the phase diagram that we obtain in Fig. 3(a). Thus, both the RZMs and MZMs are robust against non-uniform loss across the NW.

# 4 Conclusions and outlook

To summarize, in this work, we study the effect of dissipation on a 1D Rashba NW placed in direct proximity to a conventional spin-singlet $s$-wave superconductor. The effect of dissipation, including quantum jumps, is encompassed in the Liouvillian superoperator. We utilize third quantization to obtain an NH description of the system. We show that the MZMs that emerge in the topological regime of the isolated system, remain even in the dissipative regime. However, the MZMs now have a finite lifetime due to dissipation. We further discover that it is easy to engineer two kinds of ZMs starting from a non-topological system entirely due to dissipation: four RZMs and two MZMs. The RZMs are located at the two ends of the NW, however, they are not associated with any bulk states or a non-trivial bulk topology but rather originate from EPs induced by the dissipation. In contrast, the MZMs are linked to the bulk states, manifested both by a topological phase transition where the bulk gap closes and a finite bulk winding number, computed based on pseudo-anti-Hermiticity symmetry. We show that both the MZMs and the RZMs are robust against random disorder, but notably, the RZMs are more robust against the disorder compared to the MZMs.

For the dissipation-driven MZMs generated even when the isolated system is topologically trivial, an exciting future avenue is to showcase how the properties of the setup and the MZMs can be successfully utilized within quantum computation [39, 42–49], thereby opening for more flexible and environment-tolerant quantum computing platforms. In the case of dissipation-generated RZMs via EPs, they unravel the intricate behavior of NH systems, while also shedding light on the interplay between the NH terms and end states, beyond a NH skin effect. It would here be interesting to further investigate the topological properties of these RZMs, tied to the EPs. In particular, if it is possible to gain more control over the emergence of these modes a priori, with the knowledge of the isolated system's Hamiltonian and the specific form of the dissipation and also connect the emergence of RZMs to EPs via some analytical expressions. A next step could be to find a possibility to seek EP-generated boundary states in higher dimensions. Interestingly, the MZMs and the RZMs never overlap in parameter space. Still, having them exist side-by-side at only slightly different dissipation strengths raises questions on how to yet again distinguish MZMs from other spurious zero-energy modes. In fact, our EP-generated RZMs might be relevant to consider in current experiments seemingly not being able to detect MZMs. The form of loss may also be possible to adapt and it would be intriguing to find the form of loss by considering different types of coupling with the environment.

Furthermore, we always obtain boundary modes with a finite lifetime, which will decay over time. However, in a dissipative setup, it has been demonstrated that one can also obtain boundary modes that can have a zero or a very tiny imaginary part, implying that these states are not as affected as the remaining bulk states [70,71]. Thus, only the boundary states survive in the long time limit. A fascinating direction would be to search for such more immune MZMs, possibly even having a lifetime approaching infinity, in the presence of dissipation. In our study, we only investigate the spectral topological properties of the Liouvillian, and a next step can also be to study the topological properties of the steady state.

Finally, in our study, we only demonstrate the generation of MZMs by considering a static system. However, it is known that MZMs can also be generated from static topologically trivial systems by employing periodic drives [68,96]. Thus, it would be intriguing to investigate the interplay of periodic drive and dissipation to generate MZMs. In addition, we may ask whether we can get also RZMs in a driven dissipative system and it would be intriguing if one could even get RZMs at the Floquet zone boundary.

## Acknowledgements

We acknowledge fruitful discussions with Rodrigo Arouca.

**Funding information**  We acknowledge financial support from the Knut and Alice Wallenberg Foundation, through the Wallenberg Academy Fellows program KAW 2019.0309 and project grant KAW 2019.0068. Part of the computations were performed at the Uppsala Multidisciplinary Center for Advanced Computational Science (UPPMAX) provided by the National Academic Infrastructure for Supercomputing in Sweden (NAISS), partially funded by the Swedish Research Council through grant agreements no. 2022-06725 and no. 2018-05973.

## A  Obtaining $\mathcal{L}_+$

In this Appendix A we discuss the procedure to obtain the matrix form of $\mathcal{L}_+$ [Eq. (10)] using the adjoint fermion representation [102–104]. Following the operations of the adjoint

fermions on the basis states $|P_{\underline{\alpha}}\rangle$, see Eq. (9), we identify the following relations

$$|w_a P_{\underline{\alpha}}\rangle = \left(\phi_a^\dagger + \phi_a\right)|P_{\underline{\alpha}}\rangle$$
$$|P_{\underline{\alpha}} w_a\rangle = \mathcal{P}_{\mathrm{F}}\left(\phi_a^\dagger - \phi_a\right)|P_{\underline{\alpha}}\rangle \,, \tag{A.1}$$

where the last equation is obtained employing the anti-commutation properties of the Majorana operators $w_a$. Here $\mathcal{P}_{\mathrm{F}}$ is the fermion parity operator defined as $\mathcal{P}_{\mathrm{F}} = (-1)^{\sum_a \alpha_a} = \exp(i\pi\mathcal{N})$, with $\mathcal{N}$ being the number operator. Employing Eqs. (A.1) and the adjoint fermion anti-commutation relations, we obtain the following identities, which are useful to find the quadratic representation of the Liouvillian:

$$|w_a w_b P_{\underline{\alpha}}\rangle = \left(\phi_a^\dagger \phi_b^\dagger + \phi_a^\dagger \phi_b + \phi_a \phi_b^\dagger + \phi_a \phi_b\right)|P_{\underline{\alpha}}\rangle \,, \tag{A.2a}$$

$$|P_{\underline{\alpha}} w_a w_b\rangle = \left(\phi_a^\dagger \phi_b^\dagger - \phi_a^\dagger \phi_b - \phi_a \phi_b^\dagger + \phi_a \phi_b + 2\delta_{a,b}\right)|P_{\underline{\alpha}}\rangle \,, \tag{A.2b}$$

$$|w_a P_{\underline{\alpha}} w_b\rangle = \mathcal{P}_{\mathrm{F}}\left(\phi_a^\dagger \phi_b^\dagger - \phi_a^\dagger \phi_b + \phi_a \phi_b^\dagger - \phi_a \phi_b\right)|P_{\underline{\alpha}}\rangle \,. \tag{A.2c}$$

The density matrix $\rho(t)$ can now be represented by the basis element $|P_{\underline{\alpha}}\rangle$. The unitary part of the Liouvillian $\mathcal{L}$ reads as

$$\begin{aligned}
\mathcal{L}_0 \rho &:= -i\left[H, \rho(t)\right] \\
&= -i\sum_{a,b} H_{a,b}\left(|w_a w_b P_{\underline{\alpha}}\rangle - |P_{\underline{\alpha}} w_a w_b\rangle\right) \\
&= -4i\sum_{a,b} \phi_a^\dagger H_{a,b} \phi_b \quad \left(\text{using Eqs. (A.2a), (A.2b), and } H_{\mathrm{M}}^{ba} = -H_{\mathrm{M}}^{ab}\right) \\
&= -4i\underline{\phi}^\dagger \cdot H_{\mathrm{M}} \underline{\phi} \,, \tag{A.3}
\end{aligned}$$

while the non-unitary part can be represented as

$$\begin{aligned}
\mathcal{D}[L_m]\rho &:= -\frac{1}{2}\sum_m \left(\{L_m^\dagger L_m, \rho(t)\} - 2L_m^\dagger \rho(t) L_m^\dagger\right) \\
&= -\frac{1}{2}\sum_{m,a,b} l_{m,a} l_{m,b}^* \left(|w_a w_b P_{\underline{\alpha}}\rangle + |P_{\underline{\alpha}} w_a w_b\rangle - 2|w_a P_{\underline{\alpha}} w_b\rangle\right) \\
&= \sum_{m,a,b} \frac{1+\mathcal{P}_{\mathrm{F}}}{2}\left[\phi_a^\dagger \left(l_{m,a} l_{m,b}^* - l_{m,b} l_{m,b}^*\right)\phi_b^\dagger - \phi_a^\dagger \left(l_{m,a} l_{m,b}^* + l_{m,b} l_{m,b}^*\right)\phi_b\right] \\
&\quad + \sum_{m,a,b} \frac{1-\mathcal{P}_{\mathrm{F}}}{2}\left[\phi_a \left(l_{m,a} l_{m,b}^* - l_{m,b} l_{m,b}^*\right)\phi_b - \phi_a \left(l_{m,a} l_{m,b}^* + l_{m,b} l_{m,b}^*\right)\phi_b^\dagger\right] \\
&\qquad\qquad \text{(using Eqs. (A.2a), (A.2b), and (A.2c))} \\
&= \sum_{a,b} \frac{1+\mathcal{P}_{\mathrm{F}}}{2}\left[\phi_a^\dagger \left(M_{a,b} - M_{b,a}\right)\phi_b^\dagger - \phi_a^\dagger \left(M_{a,b} + M_{b,a}\right)\phi_b\right] \\
&\quad + \sum_{a,b} \frac{1-\mathcal{P}_{\mathrm{F}}}{2}\left[\phi_a \left(M_{a,b} - M_{b,a}\right)\phi_b - \phi_a \left(M_{a,b} + M_{b,a}\right)\phi_b^\dagger\right] \\
&\qquad\qquad \left(\text{defining } M_{a,b} = \sum_m l_{m,a} l_{m,b}^*\right) \\
&= \frac{1+\mathcal{P}_{\mathrm{F}}}{2}\left[\underline{\phi}^\dagger \cdot \left(M - M^T\right)\underline{\phi}^\dagger - \underline{\phi}^\dagger \cdot \left(M + M^T\right)\underline{\phi}\right] \\
&\quad + \frac{1-\mathcal{P}_{\mathrm{F}}}{2}\left[\underline{\phi} \cdot \left(M - M^T\right)\underline{\phi} - \underline{\phi} \cdot \left(M + M^T\right)\underline{\phi}^\dagger\right] \,. \tag{A.4}
\end{aligned}$$

Thus, the full Liouvillian reads as

$$\mathcal{L} = -4i\underline{\phi}^\dagger \cdot H_M \underline{\phi} + \frac{1+\mathcal{P}_F}{2}\left[\underline{\phi}^\dagger \cdot \left(M - M^T\right)\underline{\phi}^\dagger - \underline{\phi}^\dagger \cdot \left(M + M^T\right)\underline{\phi}\right]$$
$$+ \frac{1-\mathcal{P}_F}{2}\left[\underline{\phi} \cdot \left(M - M^T\right)\underline{\phi} - \underline{\phi} \cdot \left(M + M^T\right)\underline{\phi}^\dagger\right]. \tag{A.5}$$

Now, considering the physical situation where have an even number of fermionic operators, we can set $\mathcal{P}_F = +1$. In this case, the Liouvillian $\mathcal{L}$, Eq. (A.5) boils down to $\mathcal{L}_+$, Eq. (10). This concludes the derivation of $\mathcal{L}_+$.

# B  Computation of winding number

In this Appendix B, we provide the steps to obtain the definition of the winding number $\nu$ in Eq. (13). The main ingredient to compute the winding number is the pseudo-anti-Hermiticity symmetry $\Gamma$. The matrix $X$ being NH means left and right eigenstates are different:

$$X(k)|\Psi_{R,n}(k)\rangle = E_n(k)|\Psi_{R,n}(k)\rangle \quad \text{and} \quad X^\dagger(k)|\Psi_{L,n}(k)\rangle = E_n^*(k)|\Psi_{L,n}(k)\rangle. \tag{B.1}$$

The pseudo-anti-Hermiticity symmetry demands that $|\Psi_{L,-n}(k)\rangle = \Gamma|\Psi_{R,n}(k)\rangle$ such that $E_{-n}^*(k) = -E_n(k)$ [105]. Thus, we define an operator $Q(k)$ as

$$Q(k) = \frac{1}{2}\Bigg(\sum_{n>0}|\Psi_{L,n}(k)\rangle\langle\Psi_{R,n}(k)| - \sum_{n<0}|\Psi_{L,n}(k)\rangle\langle\Psi_{R,n}(k)|$$
$$+ \sum_{n>0}|\Psi_{R,n}(k)\rangle\langle\Psi_{L,n}(k)| - \sum_{n<0}|\Psi_{R,n}(k)\rangle\langle\Psi_{L,n}(k)|\Bigg). \tag{B.2}$$

Here, $Q(k)$ is a Hermitian matrix satisfying $\Gamma Q(k)\Gamma = -Q(k)$ [105]. Thus, $\Gamma$ appears as an emergent chiral symmetry for $Q(k)$, although $X(k)$ explicitly breaks it. Employing the basis in which $\Gamma$ is diagonal with $U_\Gamma \Gamma U_\Gamma^{-1} = \text{diag}(1,1,-1,-1)$, we can represent $Q(k)$ as

$$U_\Gamma Q(k)U_\Gamma^{-1} = \begin{pmatrix} 0 & q(k) \\ q^*(k) & 0 \end{pmatrix}. \tag{B.3}$$

Using $q(k)$, we define the topological invariant $\nu$ for the system in Eq. (13). We can also compute this topological invariant in real space for a system obeying PBC [106].

# C  NW-SC heterostructure under single-site loss

In the main text, we consider the case where the whole NW is subjected to dissipation. However, we can also easily envision a setup where only a few or even a single lattice site is exposed to the environment and encapsulates the effect of dissipation. In this appendix, we report the results of the extreme other limit from the main text, that of single-site dissipation. Overall, we find similar results as in the main text, except that we also often obtain states at the actual dissipation site. As a consequence, we report these results as an appendix, as they primarily complement the main text results. Akin to the main text, we divide our discussions into two parts: the effect of dissipation on a topological and non-topological isolated NW-SC heterostructure.

Overall, we discuss the two limiting cases: loss at one of the middle sites of the chain, here at $n = 50$, and loss at the very end site of the chain, $n = 1$, considering a NW with 100

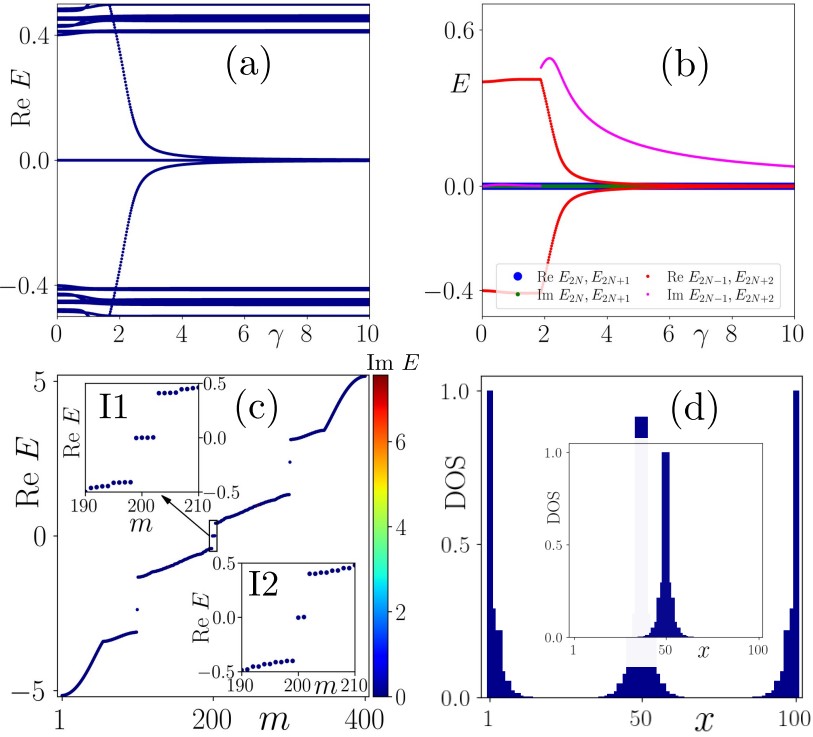

Figure 7: (a) Real part of the Lindblad spectra Re $E$ as a function of dissipation strength $\gamma$. (b) Eigenvalues for the four middle states. Both the real and imaginary parts of eigenvalue for the $2N$- and $(2N+1)$-th states stay at zero, while the $(2N-1)$-th and $(2N-2)$-th state show some transition. (c) Eigenvalues as a function of state index, employing OBC for $\gamma = 8$. The zoomed-in eigenvalues close to Re $E = 0$ are shown in inset I1, while inset I2 illustrates the eigenvalues for a system obeying PBC. (d) LDOS for states with Re $E = 0$. The LDOS for a system obeying PBC is shown in the inset. Here $B_x = 2.0 > |B_{x,c1}|$.

lattice sites with $n$ representing the lattice site subjected to loss. The loss at lattice site $n$ can be encapsulated via the jump operator

$$L_l^n = \sqrt{\gamma}\left(c_{n\uparrow} + c_{n\downarrow}\right), \tag{C.1}$$

using $\gamma_i = \gamma$ when $i = n$. Following this jump operator $L_l^n$, we construct the corresponding Liouvillian $\mathcal{L}_+$ and the matrix $X$, just as the main text.

## C.1 Fate of MZMs under single-site dissipation

We first consider the loss at a single site of a topologically non-trivial NW-SC heterostructure with MZMs in the Hermitian (isolated) system regime. Overall, we find that the lattice site at which the loss is implemented can significantly change its behavior, as detailed below.

### C.1.1 Dissipation in middle of NW

We first showcase the results for a system encompassing loss at one of the middle sites ($n = 50$) of the chain in Fig. 7. We plot the real part of the Lindblad spectra Re $E$ as a function of the amplitude of the loss $\gamma$ for a system obeying OBC in Fig. 7(a). We observe the presence of the MZMs at Re $E = 0$ for $\gamma = 0$, as expected. However, as we increase the strength of the dissipation, we notice that two other states come closer to the Re $E = 0$ (around $\gamma \sim 4$) and

then remain there with further increase in $\gamma$. To investigate these states more, we depict the real and imaginary parts of the eigenvalues of the four middle states, $(2N-1)$, $2N$, $(2N+1)$, and $(2N+2)$, in Fig. 7(b). Both the real (blue dots) and imaginary (green dots) parts of the eigenvalues of the MZMs stay at zero for all the values of $\gamma$. While for the $(2N-1)$-th and $(2N+2)$-th states, the real parts decrease to zero around $\gamma = 4$ (red dots), although they still have finite imaginary part (magenta dots).

Next, we consider a cut from Fig. 7(a) at $\gamma = 8.0$ and plot the corresponding real part of the Lindblad spectrum Re $E$ as a function of the state index $m$ in Fig. 7(c), with the zoomed-in eigenvalue close to Re $E = 0$ shown in inset I1. We identify the existence of four modes at Re $E = 0$ as expected from Figs. 7(a,b). In order to identify whether all the states are end modes, we further employ PBC, i.e. connect the two ends of the chain, and plot the corresponding eigenvalue spectrum in inset I2. We observe that then there exist still two modes at Re $E = 0$ for a system obeying PBC. Thus, two of the four states appearing at Re $E = 0$ are not end states, but actually exist in the bulk. To investigate more specifically the localization properties of all Re $E = 0$ modes, we compute the LDOS at Re $E = 0$ as a function of the lattice site $x$ in Fig. 7(d). We observe three peaks: two at the ends of the chains, which we associate with the MZMs, present already in the Hermitian limit, and one at the site at which the loss is incorporated. We also demonstrate the LDOS employing PBC in which only the ZMs at the loss site survive, see inset in Fig. 7(d). Thus, we conclude that strong single-site loss in the interior of the NW induces states at Re $E = 0$, localized at the loss site. This phenomenon is seemingly akin to that of defect-induced states [108,109] or the disorder-induced states in a Rashba NW [33,36], but here driven purely by dissipation.

### C.1.2 Dissipation at end site of NW

Having demonstrated the effect of loss in the interior of the NW, we now discuss the effect of the loss at one of the end sites (here $n = 1$) of the NW. In Fig. 8(a), we illustrate the real part of the Lindblad spectra Re $E$ as a function of $\gamma$. The MZMs that are present in the Hermitian (isolated) system continue to appear at Re $E = 0$. However, when we investigate the imaginary parts In $E$ in Fig. 8(b), we notice that one of the MZMs incurs a finite imaginary part for non-zero $\gamma$. Thus, one MZM suffers from having a finite lifetime. The increase in the imaginary part of the MZMs is, however, not monotonic with respect to $\gamma$, as seen in Fig. 8(b), with the imaginary part even starting to decrease for $\gamma \sim 2$, before saturating to a finite value for large $\gamma$. This is quite an interesting behavior as the MZM then becomes more resistant to dissipation with higher dissipation. The other MZM always remains unaffected by dissipation, which is expected, since it is located at the other end of the NW, which is far away from the loss site.

Next, we consider a cut from Fig. 8(a) at $\gamma = 2$ and plot the real part of the Lindblad spectrum Re $E$ as a function of state index $m$ in Fig. 8(c). The associated imaginary parts of the eigenvalue spectrum Im $E$ are indicated by color. In the inset of Fig. 8(c) we showcase the eigenvalues close to Re $E = 0$. We clearly identify the existence of two MZMs, with one getting a large finite imaginary part. To investigate the localization property of these MZMs, we plot the LDOS of the Re $E$ states as a function of lattice sites in Fig. 8(d). We observe that the MZMs are sharply localized at the ends of the system. However, the peak height of the left MZM, where loss is applied, is less compared to the right one. Also, the position of the left peak is shifted by one lattice site to the right. We observe that this shifting of the peak happens only after the imaginary part of one of the MZMs reaches the maximum value around $\gamma = 2$. Thus, the MZM counterbalances the effect of the dissipation by shifting away from the dissipative site. This explains why the imaginary eigenvalue part can decrease despite increasing dissipation. This is an interesting finding, which also signifies the robustness of MZM against dissipation. It would be intriguing to analytically find an expression that has a dependence on the position of loss on MZMs. However, we leave this for future studies.

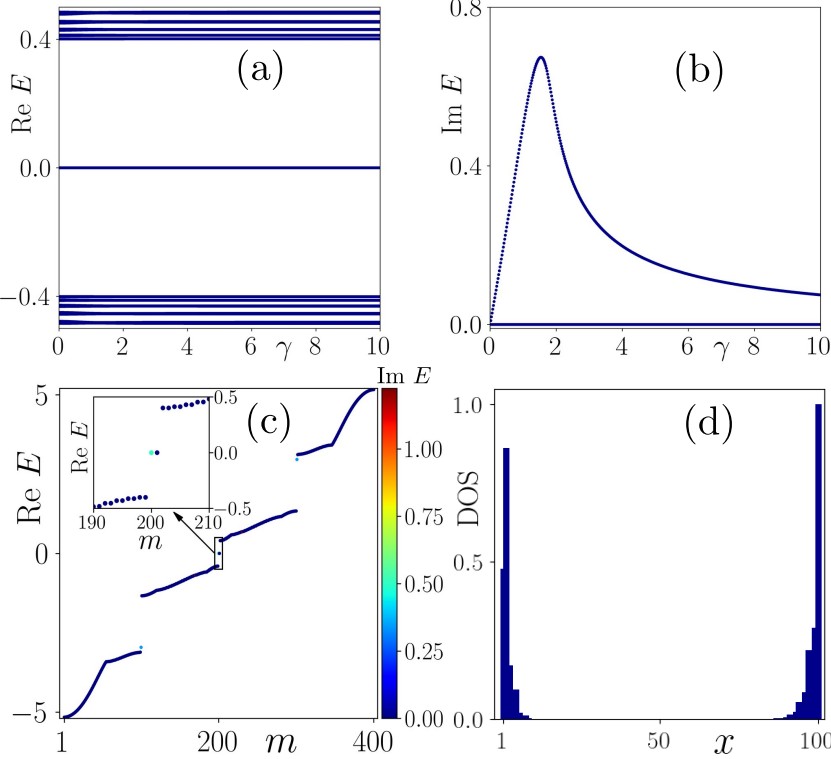

Figure 8: (a) Real part of the Lindblad spectra Re $E$ as a function of dissipation strength $\gamma$. (b) Im E for the two middle states as a function of $\gamma$. (c) Lindblad spectrum as a function of state index $m$ for a fixed $\gamma = 2$ and (d) LDOS computed for the Re $E = 0$ states. Here $B_x = 2.0 > |B_{x,c1}|$.

## C.2 Non-topological system under single-site dissipation

Having investigated the effect of single-site loss on a topological Rashba NW-SC heterostructure, we finally discuss the effect of single-site loss in a non-topological setup. We only show the results for loss at $n = 1$, since, we observe the most interesting behavior in this. Dissipation at the interior of the NW does not change the physics substantially.

### C.2.1 Dissipation at end site of NW

With the loss at one of the end sites of chain ($n = 1$), we plot the real part of the Lindblad spectra Re $E$ as a function of $\gamma$ in Fig. 9(a). We identify the emergence of ZMs at Re $E = 0$ for a range of $\gamma$. However, akin to the RZMs in Fig. 3(a) in the main text, also here, these ZMs are not associated with any bulk gap closing and can thus not have a topological bulk winding associated with them either. To understand their origin, we plot the eigenvalues, both real part Re $E$ (red) and absolute value Abs $E$ (blue), of the two middle states: $2N$ and $(2N + 1)$ in Fig. 9(b). We observe that when the modes at Re $E = 0$ appear or disappear, their absolute part Abs $E$ also merges. We also illustrate the scalar product (green) of the $2N$-th and $(2N + 1)$-th eigenstate on the right axis of Fig. 9(b). One can see that these eigenstates become parallel exactly when the corresponding absolute part of the eigenvalues becomes degenerate. Thus, we have EPs at these two points, marked with black dashed lines. We can thus relate the appearance/disappearance of ZMs to EPs. This phenomenon is similar to the RZMs that we discuss in Fig. 3(a) in the main text. However, for uniform dissipation, we find that the RZMs disappear with the increase in $\gamma$ due to the closing of the bulk gap, while the disappearance of the ZMs for single-site loss is tied to an EP.

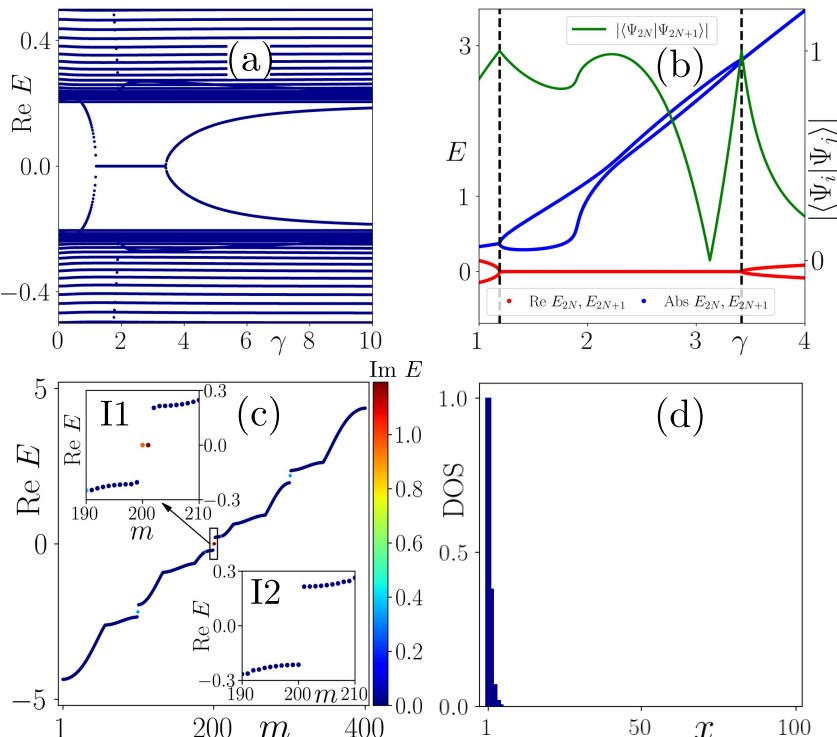

Figure 9: (a) Real part of Lindblad spectra Re $E$ as a function of $\gamma$ for a system having loss at one of the ends of the NW. (b) Eigenvalues, real Re $E$ (red dots) and absolute Abs $E$ (blue dots) for the two middle states. The scalar product of these two states $|\langle\Psi_i|\Psi_j\rangle|$ (green dots) on the right axis of (b). (c) Eigenvalue as a function of the state index, employing OBC for $\gamma = 2$. The zoomed-in eigenvalues close to Re E are shown in inset I1, while inset I2 illustrates the eigenvalues for a system obeying PBC. (d) LDOS corresponding to Re E = 0. Here $B_x = 1.2 < |B_{x,c1}|$.

In Fig. 9(c), we show the real part of the Lindblad spectra Re $E$ with their imaginary parts Im $E$ weighted by color, as a function of state index $m$. In inset I1, we depict the states close to Re $E = 0$, while in inset I2, we show the eigenvalue spectra for a system obeying PBC. We observe the presence of two states at Re $E = 0$ in the inset I1 of Fig. 9(c), but no states appearing at Re $E = 0$ in the inset I2 of Fig. 9(c). Thus, these zero-energy states appear only when the ends of the chains are open. We also demonstrate the LDOS associated with the Re $E = 0$ states in Fig. 9(d). These ZMs are localized at only one end of the system, where the loss is incorporated. This generation of ZMs by single-site loss is interesting since they appear as end states and thereby produce a signature in the zero-bias conductance peak mimicking an MZM.

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
