# Peer review of "Majorana zero-modes in a dissipative Rashba nanowire"

_SciPost Physics, doi:SciPost Phys. 17, 036 (2024)_

## Round 1 · Referee Report · Tobias Meng (Referee 1) · 2024-4-2

Strengths

1 - Detailed analysis of Majorana zero modes in quantum wires subject to dissipation
2 - Intriguing and highly relevant findings: dissipation-induced Majorana zero modes, additional robust zero modes
3 - Clear presentation of the results

Weaknesses

1 - The results have not been fully exploited, some questions that can probably be answered with what the authors have found remain open
2 - Implications for Majoranas as potential topological qubits unclear

Report

In their manuscript, the authors discuss a 1D topological superconductor subject to dissipation. In particular, they analyze the presence of edge-localised zero modes in the spectrum of the Liouvillian.

They find several very interesting results. Majorana zero modes of the decoupled (hermitian) model remain robust under dissipation. Dissipation larger than a critical value can stabilize Majorana zero modes even in a topologically trivial regime (as indicated by a bulk gap closing and a change in winding number). Furthermore, additional non-topological zero modes can be induced in the trivial regime via exceptional points. Both Majorana and trivial zero modes are robust to on-site disorder.

These results are timely, interesting to researchers analyzing non-Hermitian topology, open quantum systems, and ones trying to create topological qubits. The presentation of the results is mostly very clear and easily readable. Given the novelty and interest of the results and the target group, I believe that the paper could eventually be published in SciPost Physics.

Before I can make that recommendation, however, I have a number of points that I would kindly ask the authors to consider and address.

Requested changes

  1. Maybe most importantly, the interpretation of the results should be sharpened. More precisely, what do the zero modes mean physically? To elaborate: in a hermitian setting, Majorana zero modes (MZMs) for example encode a ground state degeneracy due to a spatially split complex fermionic state, which in turn can be used for potentially robust qubits. Indeed, the authors state that the MZMs acquire a finite lifetime set by the dissipation \gamma. While the authors do not explicitly comment on it, this probably implies all information stored in dissipative MZMs to be effectively washed out within the time scale set by \gamma.

However, there is probably more to the story that is worth mentioning. The MZMs relate to edge states of the damping matrix X (note that the spectrum of the Liouvillian is governed by the damping matrix). As the authors correctly point out, the Liouvillian governs the time-evolution of the density matrix. But what is the steady state that the system evolves to? Which physical observable can be connected to the edge Majoranas (or the „robust zero modes“ [RZMs]), and on which time scales (In the steady state? In the asymptotic approach towards the steady state? In an initial time range?)? The authors for example write at the end of Sec. 3.1 that the MZMs survive - yes, but what is the physical quantity related to them that survives?

Less important, but still to be discussed:

  1. The loss specified in Eq. (4) is definitely not a generic loss: it can be seen as loss of electrons with eigenvalue +1 of spin-sigma_x. This is the same spin component to which the magnetic field couples. Which of the results obtained are special to this type of loss, which are generic? It would be nice to have some comment on other effect of other spin polarisations of the loss (although there remains so much to be explore about truly generic couplings to environments that additional future studies seem part of the full answer).
  2. In Fig. 3 (a), the MZMs seem show a splitting at large disorder. Are they split at all disorder strengths (maybe just very weakly so)? Is the splitting merely due to an increased overlap of the Majorana wavefunctions with disorder?
  3. In Sec. 3.2.1, disorder averages are performed. The authors should comment a bit more on what exactly they do. Should I think of the data shown in Fig. 5 as running different microscopic disorder configurations, then ordering the states in some form (e.g. the real part of their energy or so), and then averaging the eigenvalues with same ordering number over the disorder configurations? Also, it would have been easier for me to have the details of disorder averaging (e.g. 50 runs) in the main text, not the figure caption.
  4. Maybe related to point 3, why do the averaged RZMs in Fig. 6(a) all have identical imaginary parts? If one looks at individual disorder runs, do these states still come in pairs with identical imaginary parts within the pair?
  5. Grouping RZMs into pairs with identical imaginary parts, do both pairs have weight at both ends, or is one pair located at one end, and the other pair at the other end (if that were the case, why the asymmetry)?
  6. Can the authors confirm that the disorder averaging converges after 50 configurations?
  7. Do the authors have any idea as to why the peaks of the end states are sometimes not right-left-symmetric? Is that a feature that has converged w.r.t. disorder configurations? Is only the maximal peak height different, but the integrated weight per side remains the same (which would be an edge-dependent smearing out)? Is there a shift of weight from one side to the other?
  8. In Fig. 2 (b), 4(c), 6(c), the dots for the MZMs are all red. Red is the end of the shown color scale for the imaginary part of the eigenvalue. Are the MZMs modes with maximal imaginary part, or are they just modes of „high“ imaginary part?
  9. Can the authors say more about the robust zero modes (RZMs)? Could they for example identify their wave functions or energies analytically? Is there an analytical way to connect them to the exceptional points (the numerics are certainly quite convincing, but maybe that could help identify the reason for their robustness)?

Finally, I noticed a couple of minor issues - nothing dramatic, but let me just point them out.

  1. The authors use the formulation that the density matrix decays - that is a bit ambiguous. The density matrix preserves its trace. It is correct, however, that it evolves towards its steady state expression with an exponential time-dependence.
  2. In Eq. (1), the „+h.c.“ leads to a doubling of the hermitian terms (chemical potential, magnetic field). One could introduce an extra factor 1/2, or add the +h.c. only for the terms that need it.
  3. To be overly picky, the formulation „we consider uniform loss i.e., L_i \neq o \forall i“ below Eq. (4) would be more on point if the loss amplitude in Eq. (4) would be \gamma_i, and then one could set \gamma_i=\gamma in the main text and \gamma_i spatially-dependent as chosen in the Appendix

  • validity: top
  • significance: high
  • originality: good
  • clarity: top
  • formatting: perfect
  • grammar: perfect

Author:  Arnob Kumar Ghosh  on 2024-04-16  [id 4426]

(in reply to Report 1 by Tobias Meng on 2024-04-02)
Category:
answer to question

We thank the referee for carefully reading our manuscript and the overall positive report. We provide our response to the questions raised by the referee in the attached file.

Attachment:

Report1.pdf

---

## Round 1 · Referee Report · Anonymous (Referee 2) · 2024-5-7

Strengths

  1. Clear presentation and introduction into the subject.
  2. Interesting and important findings.
  3. Relevant regimes in them model are discussed.

Weaknesses

  1. Uniform dissipation seems a bit unrealistic.

Report

In this paper, the stability of Majorana zero modes against dissipation is studied in a 1D topological superconducting model consisting of a 1D nanowire with Rashba SOC placed on top of an s-wave SC with an applied in-plane magnetic field. The authors assume uniform loss on each site of the system, and solve the full Lindblad master equation using third quantisation. They show that in the topological regime, the Majorana zero modes are robust against dissipation. Interestingly, in the topologically trivial regime, new types of zero modes appear, which are not topologically protected, whereas the dissipation can also drive a topological phase transition such that Majorana zero modes also appear.

In my opinion, this paper certainly meets SciPost's acceptance criteria, and is well suited to this journal. In particular, I believe this work opens up a new pathway in an already existing research direction in that it finds new surprising results regarding the dissipation-induced appearance of Majorana modes.

Requested changes

  1. I am a bit confused about the following statement on page 5: "This adds substantial complexity to solving the problem, and effects have also been observed in experiments", which refers to the inclusion of quantum jumps. I find the second part of this statement somewhat vague. What kind of effects are the authors referring to?

  2. Above Eq.(4), the authors state "For simplicity, we consider a simple form of the onsite loss, which can become experimentally feasible." Could the authors comment on this? This also relates to my next point.

  3. In Eq.(4) the authors introduce the jump operator they use for their model. In their choice, they assume uniform loss in the entire system. Would it not be more realistic to have different loss rates on different lattice sites? Could the authors comment on how that would alter their results?

  4. Below Eq.(12), the authors state that \tilde{\tau} and \tilde{\sigma} are newly defined Pauli matrices. For the sake of completion, could they add the explicit form of these new Pauli matrices to the appendix?

  5. In the beginning of section 2.3, the authors introduce a symmetry they call pseudo-anti-Hermiticity symmetry. I want to note that within the framework introduced by Kawabata et al. (PRX 9, 041015 (2019)) this symmetry is referred to as chiral symmetry. It may be worthwhile pointing out that this symmetry appears under a different name as well.

  6. The labels and texts in the insets in the figures are very hard to read because they are very small. For example, it is very difficult to read the labels in the inset in Fig.2(b) or to read the legend in Fig.3(b).

  7. For the model discussed in Figure 3, the authors find four so-called RZMs (robust zero-energy modes) induced by two second-order EPs (EP2s) on the left side, i.e., for gamma a bit larger than one. It is known that EP2s always come in pairs and are connected via so-called (i-)Fermi arcs, i.e., via branch cuts at which the real (imaginary) part of the energy is degenerate. This behaviour is also visible in Fig.3(b) for the red and black curves in the bottom. As such, the appearance of the RZMs to me looks like the Fermi arcs one would expect to see between EP2s. Is this indeed a correct observation? If so, the set of Fermi arcs, which would amount to a four-fold degeneracy in this case in line with the observation that four RZMs exist, must terminate at another set of EP2s. Do such EP2s appear at the bulk-gap closing points? Am I correct when the imaginary part of the energy also disappears for the RZMs? Also, does the symmetry in the model force the EP2s and the RZMs to sit at zero energy? Seeing that there is a double set of EP2s, I would expect they could in principle sit away from zero energy as long as they preserve the spectral symmetry, i.e., appear as epsilon and -epsilon. Could there be a scenario where the EP2s and RZMs could move away from zero?

  8. The dark stars in Figs.3(d) and (e) in the red regions are very hard to spot. Could the authors choose a different colour?

  9. In the appendix, the authors study the very interesting case of having single-site losses. I believe some of these results may need to featured in the main text because they are quite fascinating.

Recommendation

Ask for minor revision

  • validity: high
  • significance: high
  • originality: high
  • clarity: high
  • formatting: excellent
  • grammar: excellent

Author:  Arnob Kumar Ghosh  on 2024-05-20  [id 4495]

(in reply to Report 2 on 2024-05-07)
Category:
answer to question

We thank the referee for carefully reading our manuscript and recommending it for publication. We respond to the questions raised by the referee in the attached file.

Attachment:

Report2.pdf

---

## Round 2 · Referee Report · Tobias Meng (Referee 1) · 2024-5-31

Strengths

1 - Detailed analysis of Majorana zero modes in quantum wires subject to dissipation
2 - Intriguing and highly relevant findings: dissipation-induced Majorana zero modes, additional robust zero modes
3 - Clear presentation of the results

Report

In their resubmission and the accompanying response letter, the authors have provided careful explanations of the questions I had, and made reasonable corresponding changes to the manuscript. The same applies, from my standpoint, to the questions raised by the second referee.

As mentioned in my first report, the results presented in this paper are timely, interesting to researchers analyzing non-Hermitian topology, open quantum systems, and ones trying to create topological qubits. I therefore beliebe that the manuscript meets the criteria for acceptance in SciPost Physics, and recommend its publication in the present form.

Recommendation

Publish (easily meets expectations and criteria for this Journal; among top 50%)

---

## Round 2 · Author Response

We thank the editor for taking care of our paper and considering it for the review process. We thank both referees for carefully reading our manuscript, providing critical comments, and overall positive reports. We have provided the answers to questions raised by both the referees as individual author replies.

---

## Round 2 · List of Changes

• Corrected typo in Eq. (1).
  • Added/modified the sentences in Section 2.2: "This adds substantial complexity... necessary for open systems."
  • Changed $\gamma$ to $\gamma_i$ in Eq. (4).
  • Added/modified the sentences at the end of Section 2.2.1: "... and approaches its steady state... 38-fold classification instead [66]."
  • Added a sentence in Section 2.3: "For an NH system, this symmetry is also called chiral symmetry [66]."
  • Added/modified the sentences in Section 3.1: "... and as such they would appear... with the maximum imaginary parts." and "However, the MZMs now also carry... ZMs we obtain in this work."
  • Changed the colormap in Figs. 3(d,e).
  • Increased the font size of the labels and the texts in the insets of all the figures.
  • Added a few sentences in Section 3.2: "We finally note that, in Fig. 3(a,c)... disappears for longer NWs." and "Furthermore, we observe that... localized at two opposite ends." and "Furthermore, we also investigate... dissipation-induced MZMs and RZMs."
  • Added a few sentences in Section 3.2.1: "We consider disorder averages over 50 and 500... corresponding imaginary parts in ascending order." and also "We also investigate the effect of non-uniform dissipation... robust against non-uniform loss across the NW."
  • Added/modified the sentences in Section 4: "... and also connect the emergence of RZMs to EPs via some analytical expressions." and "The form of loss may also be... coupling with the environment." and also "In our study, we only ... topological properties of the steady state."

---

## Editorial Decision

published